# Relationship between baseline platelet-to-red blood cell distribution width ratio and all-cause mortality in non-traumatic subarachnoid hemorrhage: A retrospective analysis of the MIMIC-IV database

Yang Liu[1], Yi Shi[2], Pengzhao Zhang[3], Mengyuan Xu[1], Jiaqi Zhang[2], Jing Xia[3], Shaojie Guo[3], Gaofeng Li[4], Guang Feng[4]*

1 Department of Neurosurgical ICU, Zhengzhou University People's Hospital, Henan Provincial People's Hospital, Zhengzhou, China, 2 Department of Neurosurgical ICU, Henan University People's Hospital, Henan Provincial People's Hospital, Zhengzhou, China, 3 Graduate School of Xinxiang Medical University, Xinxiang, China, 4 Department of Neurosurgical ICU, Henan Provincial People's Hospital, Zhengzhou, China

* 19903711799@163.com

## Abstract

### Objective

The study aimed to evaluate the relationship between baseline platelet-to-red blood cell distribution width ratio (PRR) and mortality in critically ill patients with non-traumatic subarachnoid hemorrhage (SAH).

### Methods

This cohort study of adults with non-traumatic SAH used Medical Information Mart for Intensive Care (MIMIC-IV) data from 2008–2022 admissions at the Intensive Care Unit (ICU). We collected the PRR levels at admission and determined the all-cause death rates for the ICU and hospital. Cox proportional hazards models were utilized to analyze the association between baseline PRR level and all-cause mortality. Kaplan–Meier survival curve analysis was used to examine the consistency of these correlations. Restricted Cubic Splines (RCS) analysis was used to determine the relationship curve between all-cause mortality and PRR level and examine the threshold saturation effect. To evaluate the consistency of correlations, interaction and subgroup analyses were also conducted.

### Results

A total of 1056 patients with non-traumatic SAH were included in this study. All-cause mortalities in the ICU and hospital were 14.8% (156/1056) and 18.6% (196/1056), respectively. Compared to individuals with lower PRR Q1(≤12.67), the adjusted

**Data availability statement:** All data files are available from the MIMIC-IV database. https://physionet.org/content/mimiciv/3.0/.

**Funding:** This work was supported by the Health Commission of Henan Province (grant number: SBGJ202402008). The funders had no role in study design, data collection and analysis, decision to publish, or preparation of the manuscript. Guang Feng (the funding recipient) contributed to study design, data collection and analysis, and manuscript preparation.

**Competing interests:** The authors have declared that no competing interests exist.

HR values in Q2 (12.68–15.99), Q3 (16.00–19.41), and Q4 (≥19.42) were 0.61 (95%CI:0.40–0.92, p = 0.017), 0.60 (95%CI: 0.39–0.92, p = 0.020), and 0.60 (95% CI:0.39–0.93, p = 0.019), respectively. Kaplan–Meier analysis showed that patients with low PRR levels had significantly higher ICU and in-hospital mortality (p < 0.001). The association between the PRR level and ICU and in-hospital mortality exhibited a non-linear relationship (p < 0.05). The threshold breakpoint value of 22.6 was calculated using RCS analysis. When the PRR level was lower than 22.6, the risk of ICU and in-hospital mortality rates decreased with an HR of 0.91 (95%CI: 0.88–0.94, p < 0.001) and 0.94 (95%CI: 0.92–0.96, p < 0.001), respectively. When the PRR level was higher than 22.6, the risk of ICU mortality (HR = 1.03, 95% CI: 0.97–1.10, p = 0.312) and in-hospital mortality (HR = 1.01, 95%CI: 0.95–1.08, p = 0.693) almost hardly increased with the increase in the PRR level. The interaction between the PRR and all subgroup factors was analyzed, and significant interactions were not observed.

## Conclusion

There was a non-linear connection between the baseline PRR level and in-hospital mortality. A low level of PRR could increase the risk of death in participants with non-traumatic SAH.

---

## Introduction

Subarachnoid hemorrhage (SAH) is a stroke caused by bleeding into the subarachnoid space, the area between the arachnoid membrane and the pia mater surrounding the brain. It is characterized by sudden and severe headaches, often described as the worst headache of one's life, and can lead to significant morbidity and mortality. SAH can be classified as traumatic or non-traumatic, with non-traumatic SAH often resulting from the rupture of an aneurysm or arteriovenous malformation [1]. Epidemiological studies have shown that non-traumatic SAH accounts for approximately 5% of all strokes and has a high case fatality rate, with up to 50% of patients dying within the first 30 days of the event [2]. Survivors of non-traumatic SAH may suffer from significant neurological deficits, including cognitive impairment, weakness, and difficulty with speech or vision [3]. Current treatment strategies for non-traumatic SAH include surgical clipping or endovascular coiling of the ruptured aneurysm, along with supportive care in the intensive care unit to prevent complications such as vasospasm, cerebral ischemia, and hydrocephalus [2]. Despite advances in the management of SAH, predicting outcomes remains challenging due to the complex interplay of various factors that influence patient prognosis. There is still a need for better prognostic markers to identify individuals at high risk of poor outcomes. Recent research has highlighted the importance of early identification and management of brain injury in patients with SAH. Early brain injury, which occurs within the first 72 h after SAH, is characterized by a cascade of inflammatory responses, oxidative stress,

and neuronal apoptosis [4]. Targeting these pathological mechanisms may improve outcomes and reduce mortality in patients with non-traumatic SAH. The platelet-to-red blood cell distribution width ratio (PRR) is a novel metric that reflects the balance between platelet count and red blood cell (RBC) distribution width. However, the relationship between baseline PRR and outcomes in patients with non-traumatic SAH remains poorly understood. After SAH, platelets release substances that drive inflammation, and a higher PRR may indicate a more active platelet-mediated inflammatory response. A low PRR can result from pathological thrombocytopenia. Platelets generate reactive oxygen species (ROS), causing oxidative damage, while red blood cells have antioxidant functions; changes in PRR reflect oxidative stress imbalances. Inflammatory cytokines and oxidative stress after SAH trigger neuronal apoptosis, and platelets are involved through interactions with the blood brain barrier. PRR may serve as a biomarker for the balance between platelet-related pro-apoptotic effects and the red blood cell state. In this study, we aimed to investigate the relationship between baseline PRR and all-cause mortality in patients with non-traumatic SAH using data from the Medical Information Mart for Intensive Care IV (MIMIC-IV) database. By exploring the pathophysiological mechanisms underlying this association, we hoped to improve risk stratification and clinical management for patients with non-traumatic SAH.

## Methods

### Study population

The source of our data was the MIMIC-IV (v3.0), (Johnson, A., Bulgarelli, L., Pollard, T., Gow, B., Moody, B., Horng, S., Celi, L. A., & Mark, R. (2024). MIMIC-IV (version 3.0). PhysioNet. https://doi.org/10.13026/hxp0-hg59) large-scale, open-source database that was developed and maintained by the MIT Computational Physiology Laboratory. It included the records of all patients admitted to Beth Israel Deaconess Medical Center from 2008 to 2022. The database offered comprehensive data for each patient, such as laboratory results, vital signs, medication administration, length of stay, etc. To protect patient privacy, all personal information was replaced with random codes and anonymized; thus, we did not need patient consent or ethical approval. The PhysioNet online platform allows downloading the MIMIC-IV (v3.0) database. One of the authors, Gaofeng Li, passed the exams on "Conflict of Interest" and "Data or Sample Only Research" (ID:54026276) and completed the Collaborative Institutional Training Initiative course to access the database. The research team was authorized to use the database and extract data. The study was carried out following the Helsinki Declaration guidelines. All data were anonymized to protect patient privacy, and the need for informed consent was waived. This study followed the Strengthening the Reporting of Observational Studies in Epidemiology statement. Among them, 1077 patients with non-traumatic SAH were selected based on the record of ICD-9 code 430, and ICD-10 codes I602, I604, I606, I607, I608, I609, I6001, I6002, I6010, I6011, I6012, I6020, I6021, I6022, I6031, I6032, I6051, I6052, I6900, I6901, I69011, I69018, I69020, I69021, I69022, I69028, I69044, I69051, I69054, I69092, and I69098.

Patients who met the following criteria were included: (1) first ICU admission and first time in hospital; (2) age > 18 years. The exclusion criteria were as follows: (1) ICU patients with a length of stay <24 h and (2) participants who had missing platelet or RBC distribution width values. Finally, 1056 patients were included in this study (Fig 1). Details of missing data are shown in S1 Table.

### Data extraction

All relevant variables were taken from the medical record using Structured Query Language (SQL) with PostgreSQL in the MIMIC-IV database. On the first day of ICU admission, the following variables were retrieved from the aforementioned database: (1) demographic variables: age, gender, and race; (2) vital signs: systolic blood pressure (SBP), diastolic blood pressure (DBP), mean arterial pressure (MAP), temperature, heart rate (HR), respiratory rate (RR) and pulse oxygen saturation ($SpO_2$); (3) comorbidities: hypertension, diabetes, myocardial infarction, heart failure, malignant tumor, chronic kidney disease, cirrhosis, pneumonia, stroke history, hyperlipoidemia and sepsis; (4) treatment of aneurysm: clipping or

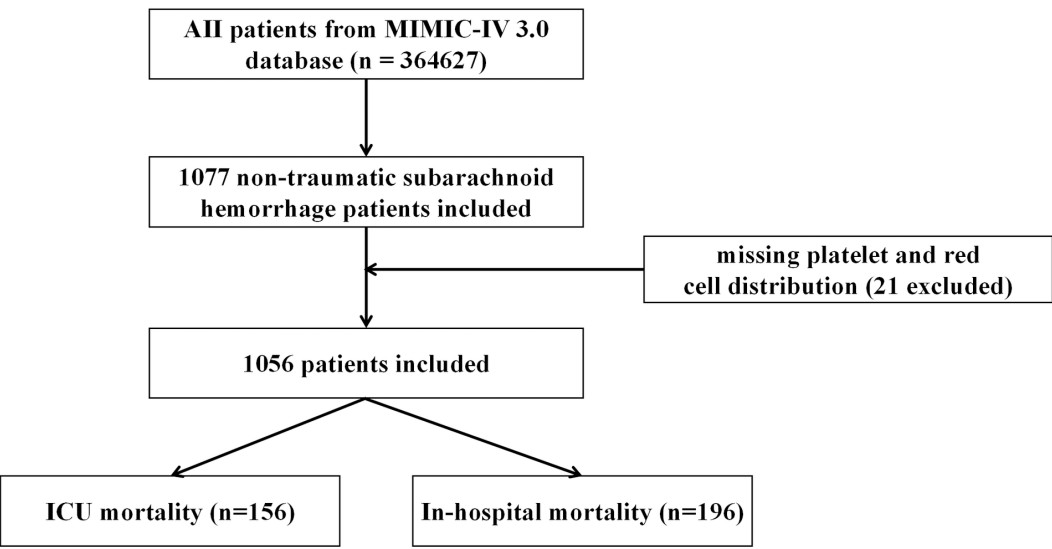

**Fig 1. Flow chart of the study.** MIMIC-IV, Medical Information Mart for Intensive Care IV; ICU, intensive care unit; SAH, subarachnoid hemorrhage.

coiling; (5) inputevents: dobutamine, dopamine, epinephrine, norepinephrine, and vasopressin; (6) mechanical ventilation; (7) within the first 24 h after ICU admission, laboratory indicators such as white blood cell count (WBC), RBC, platelet count, hemoglobin, RBC distribution width (RDW), sodium, potassium, calcium, chloride, magnesium, glucose, anion gap (AG), prothrombin time (PT), activated partial thromboplastin time (APTT), international normalized ratio (INR), urea nitrogen, and creatinine were detected; (8) scores: Glasgow Coma Scale (GCS),World Federation of Neurosurgical Societies (WFNS) grade, Simplified Acute Physiology Score II (SAPS II), and charlson comorbidity index were used to evaluate severity at the time of admission. The classification of WFNS grade was derived from the GCS score, and a GCS score of 3–12 belongs to grade IV–V, which is usually called high-grade SAH. (9) length of ICU stay (LOIS), length of hospital stay (LOHS), ICU mortality, in-hospital mortality, death within hospital 28 days, and death within ICU 28 days were all reported. PRR was calculated using the following formula: PRR = platelet (K/uL)/RDW (%).

## Endpoints

Endpoints included ICU mortality, in-hospital mortality, LOIS, and LOHS.

## Statistical analysis

The continuous variables were represented as mean, standard deviation, or median (interquartile spacing). The independent samples t-test or Mann–Whitney U test was performed depending on the normality distribution. Moreover, non-normally distributed variables were represented as the median interquartile range (IQR) and compared between groups using the Wilcoxon rank sum test. The hypothesis for categorical variables was tested using the number of cases (%) and the chi-square test (or Fisher's exact methods).

Univariate and multivariate Cox proportional hazard regression analyses were performed to reduce the interference of potential confounding factors on in-hospital mortality. The screening of confounders was based on the following criteria: [1] Certain factors may significantly impact the outcome variate based on previous experiences. [2] For univariate analysis, the variables (p < 0.05) in univariate analysis were included in multivariate Cox regression analysis. [3] Variables with

variance inflation factor (VIF) > 10 and tolerance < 0.1 were considered to have serious collinearity and were excluded. The details of the VIFs were shown in S2 Table. In the multivariate case, we performed several statistical models to ensure the stability of the results. In the crude model, no variables were adjusted. In model I, age, sex, and race were adjusted, while model II further adjusted other variables, including heart failure, malignant tumor, chronic kidney disease, cirrhosis, pneumonia, WBC, urea nitrogen, creatinine, coiling, dobutamine, norepinephrine, vasopressin, sepsis, and charlson comorbidity index. Investigation of a curve connection between baseline PRR and ICU and in-hospital mortalities was performed using restricted cubic spline (RCS) analysis. To explore the potential threshold effect of PRR on survival, the standard Cox regression model was compared to the fixed-point piecewise Cox regression model. The log-likelihood ratio test (LRT) was used to determine if the piecewise model provided a significantly better fit than the standard Cox regression model. Interaction and stratified analyses were performed for age (<60 years and ≥60 years), gender, race, hypertension, diabetes, HF, coiling, sepsis, SAPS II (<45 and ≥45), and GCS (<9 and ≥9 scores). Missing data are interpolated by k-nearest neighbors (KNN). A two-tailed p-value of <0.05 was considered statistically significant in all analyses. All statistical analyses were performed using R (version 4.2.2, R Foundation for Statistical Computing, Vienna, Austria), and MSTATA softwares (https://www.mstata.com/).

## Results

### Baseline characteristics of the participants

Fig 1 shows the enrollment flowchart based on the inclusion and exclusion criteria. Our study included 1056 participants with non-traumatic SAH with complete platelet count and RDW recorded from the MIMIC-IV database. The distribution of the baseline characteristics of the population according to baseline PRR levels in quartiles is described in Table 1 (Q1: ≤ 12.67, Q2: 12.68–15.99, Q3: 16.00–19.41, and Q4: ≥ 19.42). The demographic variables, vital signs, comorbidities, treatment of aneurysm, input..events, mechanical ventilation, laboratory variables, and scoring according to PRR levels are presented in Table 1. Statistically significant differences were observed in most laboratory values, clinical scores, and other relevant data across the groups. Participants in Q1 were generally older, with a median age of 67 years, compared to 58 years in Q4. WBC, RBC, platelet count, and several other hematological parameters showed marked variation, with increasing values observed from Q1 to Q4. For example, WBC increased from 10.0 in Q1 to 12.9 in Q4. Similarly, the SAPSIIand health conditions such as heart failure and chronic kidney disease displayed significant disparities, with higher values in Q1 and generally decreasing through to Q4. Gender distribution also differed significantly, with a higher percentage of females in Q4 (65.9%) compared to Q1 (48.5%). Furthermore, notable differences in mortality outcomes, such as in-hospital and ICU mortalities, were present across the groups, with Q1 showing higher rates of mortality compared to other groups. In S3 Table, the baseline characteristics, grouped by the in-hospital mortality variable, highlight significant differences between the two groups (p < 0.001) in terms of PRR groups. Notably, individuals who died in the hospital (non-survival group, n = 196) show a significantly higher proportion in Q1 at 44.9% compared to those who did not die in the hospital (survival group, n = 860) at 20.5%. Conversely, the distribution in Q2, Q3, and Q4 was higher in the survival group, with 26.3%, 26.7%, and 26.5% respectively, compared to the non-survival group, where these quantiles were represented by 19.4%, 17.3%, and 18.4% respectively. Patients who did not survive were generally older, with a median age of 68 years compared to 60 years for survivors (p < 0.001). Additionally, these patients exhibited higher WBC counts, lower RBC counts, and reduced platelet counts. Notably, non-survivors had elevated serum glucose levels, anion gaps, and increased blood urea nitrogen and creatinine levels. Non-survivors also had significantly higher SAPS II (p < 0.001). Certain comorbidities, such as acute renal failure (32.1% vs. 8.1%, p < 0.001), pneumonia (30.6% vs. 18.3%, p < 0.001), and heart failure (12.2% vs. 5.9%, p = 0.002), were more prevalent in non-survivors. The use of vasopressors like norepinephrine (41.8% vs. 13.7%, p < 0.001) and the presence of sepsis (70.4% vs. 43.3%, p < 0.001) were markedly higher in this group. Furthermore, non-survivors had significantly shorter hospital and ICU stays. Gender distribution showed a

**Table 1. Population characteristics by quartiles of the baseline PRR level.**

| Variables | PRR Quartiles | | | | | p-value |
|---|---|---|---|---|---|---|
| | Overall, N = 1056 | Q1(≤12.67), N = 264 | Q2(12.68–15.99), N = 264 | Q3(16.00–19.41), N = 264 | Q4(≥19.42), N = 264 | |
| **Demographic** | | | | | | |
| Age, years | 61 (51, 72) | 67 (54, 77) | 62 (52, 73) | 60 (51, 70) | 58 (48, 68) | <0.001 |
| Gender, n (%) | | | | | | <0.001 |
| Female | 599 (56.7%) | 128 (48.5%) | 134 (50.8%) | 163 (61.7%) | 174 (65.9%) | |
| Male | 457 (43.3%) | 136 (51.5%) | 130 (49.2%) | 101 (38.3%) | 90 (34.1%) | |
| **Race, n (%)** | | | | | | 0.048 |
| White | 609 (57.7%) | 128 (48.5%) | 157 (59.5%) | 158 (59.8%) | 166 (62.9%) | |
| Black | 74 (7.0%) | 24 (9.1%) | 15 (5.7%) | 15 (5.7%) | 20 (7.6%) | |
| Asian | 43 (4.1%) | 15 (5.7%) | 13 (4.9%) | 8 (3.0%) | 7 (2.7%) | |
| Other | 330 (31.3%) | 97 (36.7%) | 79 (29.9%) | 83 (31.4%) | 71 (26.9%) | |
| **Comorbidities, n (%)** | | | | | | |
| Hypertension | 535 (50.7%) | 125 (47.3%) | 138 (52.3%) | 140 (53.0%) | 132 (50.0%) | 0.558 |
| Diabetes | 216 (20.5%) | 67 (25.4%) | 56 (21.2%) | 51 (19.3%) | 42 (15.9%) | 0.055 |
| Heart failure | 75 (7.1%) | 36 (13.6%) | 17 (6.4%) | 13 (4.9%) | 9 (3.4%) | <0.001 |
| Myocardial infarction | 35 (3.3%) | 13 (4.9%) | 8 (3.0%) | 5 (1.9%) | 9 (3.4%) | 0.276 |
| Malignant tumor | 94 (8.9%) | 37 (14.0%) | 23 (8.7%) | 18 (6.8%) | 16 (6.1%) | 0.006 |
| Chronic kidney disease | 64 (6.1%) | 34 (12.9%) | 9 (3.4%) | 6 (2.3%) | 15 (5.7%) | <0.001 |
| Cirrhosis | 25 (2.4%) | 23 (8.7%) | 0 (0.0%) | 2 (0.8%) | 0 (0.0%) | <0.001 |
| Pneumonia | 217 (20.5%) | 71 (26.9%) | 52 (19.7%) | 44 (16.7%) | 50 (18.9%) | 0.024 |
| Hyperlipoidemia | 289 (27.4%) | 80 (30.3%) | 82 (31.1%) | 66 (25.0%) | 61 (23.1%) | 0.106 |
| Sepsis | 510 (48.3%) | 153 (58.0%) | 127 (48.1%) | 103 (39.0%) | 127 (48.1%) | <0.001 |
| Charlson comorbidity index | 4.0 (2.0, 6.0) | 5.0 (3.0, 7.0) | 4.0 (2.0, 5.0) | 3.0 (2.0, 5.0) | 3.0 (2.0, 5.0) | <0.001 |
| **Vital signs** | | | | | | |
| HR, beats/min | 80 (70, 91) | 82 (69, 92) | 80 (70, 91) | 78 (69, 88) | 81 (70, 94) | 0.073 |
| SBP, mmHg | 128 (114, 142) | 129 (115, 143) | 127 (112, 142) | 127 (113, 142) | 131 (117, 143) | 0.257 |
| DBP, mmHg | 71 (62, 80) | 69 (59, 78) | 71 (62, 81) | 70 (62, 80) | 72 (64, 82) | 0.017 |
| MAP, mmHg | 86 (76, 95) | 86 (75, 94) | 85 (75, 95) | 86 (76, 94) | 87 (78, 97) | 0.165 |
| RR, times/min | 17.0 (15.0, 20.0) | 18.0 (15.0, 21.3) | 17.0 (15.0, 21.0) | 17.0 (15.0, 19.0) | 17.0 (15.0, 20.0) | 0.326 |
| $SpO_2$, % | 98.0 (96.0, 100.0) | 99.0 (96.0, 100.0) | 98.0 (96.0, 100.0) | 98.0 (96.0, 100.0) | 98.0 (96.0, 100.0) | 0.884 |
| Temperature, °C | 36.8 (36.6, 37.1) | 36.8 (36.6, 37.2) | 36.9 (36.6, 37.1) | 36.8 (36.6, 37.1) | 36.8 (36.5, 37.2) | 0.810 |
| **Laboratory results** | | | | | | |
| WBC, K/uL | 11.2 (8.6, 14.5) | 10.0 (7.7, 13.4) | 11.0 (8.7, 13.5) | 11.3 (8.9, 14.2) | 12.9 (10.1, 16.5) | <0.001 |
| RBC, M/uL | 4.1 (3.7, 4.4) | 3.8 (3.3, 4.2) | 4.1 (3.7, 4.4) | 4.1 (3.7, 4.5) | 4.2 (3.8, 4.6) | <0.001 |
| Platelets, K/uL | 214 (172, 261) | 145 (111, 162) | 195 (181, 206) | 232 (219, 245) | 296 (272, 344) | <0.001 |
| Hemoglobin, g/dL | 12.3 (11.1, 13.4) | 11.4 (9.8, 12.7) | 12.6 (11.4, 13.5) | 12.5 (11.4, 13.7) | 12.5 (11.5, 13.9) | <0.001 |
| RDW, % | 13.4 (12.8, 14.3) | 14.3 (13.4, 15.7) | 13.4 (12.8, 14.0) | 13.2 (12.6, 13.7) | 13.2 (12.6, 13.8) | <0.001 |
| Sodium, mEq/L | 139.0 (137.0, 141.0) | 139.0 (137.0, 142.0) | 140.0 (138.0, 142.0) | 139.0 (137.0, 141.0) | 139.0 (137.0, 141.0) | 0.002 |
| Potassium, mEq/L | 3.9 (3.6, 4.2) | 3.8 (3.5, 4.3) | 3.9 (3.6, 4.2) | 3.9 (3.6, 4.2) | 3.9 (3.6, 4.2) | 0.733 |
| Magnesium, mg/dL | 1.9 (1.7, 2.1) | 1.9 (1.7, 2.1) | 1.9 (1.7, 2.0) | 1.9 (1.8, 2.1) | 1.9 (1.7, 2.1) | 0.547 |
| Calciumtotal, mg/dL | 8.6 (8.2, 9.0) | 8.4 (7.9, 8.8) | 8.6 (8.2, 8.9) | 8.7 (8.3, 9.0) | 8.7 (8.2, 9.1) | <0.001 |
| Chloride, mEq/L | 105.0 (102.0, 108.0) | 105.0 (102.0, 108.0) | 105.0 (103.0, 108.0) | 105.0 (102.0, 107.0) | 104.0 (101.8, 107.0) | 0.047 |
| Glucose, mg/dL | 130 (110, 157) | 130 (109, 158) | 128 (110, 155) | 131 (111, 150) | 132 (113, 162) | 0.485 |
| Anion gap, mEq/L | 14 (12, 16) | 14 (12, 16) | 14 (12, 16) | 14 (12, 16) | 14 (12, 17) | 0.274 |

*(Continued)*

**Table 1.** (Continued)

| Variables | PRR Quartiles | | | | | p-value |
|---|---|---|---|---|---|---|
| | Overall, N = 1056 | Q1(≤12.67), N = 264 | Q2(12.68–15.99), N = 264 | Q3(16.00–19.41), N = 264 | Q4(≥19.42), N = 264 | |
| PT, s | 12.5 (11.7, 13.4) | 13.0 (12.1, 15.0) | 12.3 (11.5, 13.1) | 12.3 (11.4, 13.0) | 12.4 (11.7, 13.2) | <0.001 |
| APTT, s | 28 (26, 32) | 29 (26, 33) | 28 (26, 31) | 28 (26, 31) | 28 (26, 32) | 0.005 |
| INR | 1.1 (1.1, 1.2) | 1.2 (1.1, 1.4) | 1.1 (1.1, 1.2) | 1.1 (1.0, 1.2) | 1.1 (1.1, 1.2) | <0.001 |
| Ureanitrogen, mg/dL | 14 (10, 18) | 17 (12, 24) | 13 (10, 16) | 13 (10, 16) | 12 (10, 16) | <0.001 |
| Creatinine, mg/dL | 0.8 (0.6, 1.0) | 0.9 (0.7, 1.2) | 0.8 (0.7, 0.9) | 0.8 (0.6, 0.9) | 0.7 (0.6, 0.9) | <0.001 |
| **Scores** | | | | | | |
| SAPSII | 30 (23, 39) | 36 (27, 46) | 29 (22, 38) | 28 (22, 37) | 28 (22, 38) | <0.001 |
| GCS | 14.0 (12.0, 15.0) | 14.0 (10.0, 15.0) | 14.0 (13.0, 15.0) | 14.0 (12.8, 15.0) | 14.0 (13.0, 15.0) | 0.788 |
| WFNS grade, n (%) | | | | | | 0.101 |
| Low (I-III) | 784 (74.2%) | 181 (68.6%) | 203 (76.9%) | 198 (75.0%) | 202 (76.5%) | |
| High (IV-V) | 272 (25.8%) | 83 (31.4%) | 61 (23.1%) | 66 (25.0%) | 62 (23.5%) | |
| **Therapy, n (%)** | | | | | | |
| Clipping | 35 (3.3%) | 5 (1.9%) | 8 (3.0%) | 9 (3.4%) | 13 (4.9%) | 0.276 |
| Coiling | 184 (17.4%) | 39 (14.8%) | 42 (15.9%) | 40 (15.2%) | 63 (23.9%) | 0.016 |
| Ventilation | 797 (75.5%) | 210 (79.5%) | 194 (73.5%) | 189 (71.6%) | 204 (77.3%) | 0.136 |
| Dobutamine | 9 (0.9%) | 4 (1.5%) | 0 (0.0%) | 0 (0.0%) | 5 (1.9%) | 0.014 |
| Dopamine | 15 (1.4%) | 5 (1.9%) | 2 (0.8%) | 2 (0.8%) | 6 (2.3%) | 0.389 |
| Epinephrine | 26 (2.5%) | 9 (3.4%) | 4 (1.5%) | 5 (1.9%) | 8 (3.0%) | 0.443 |
| Norepinephrine | 200 (18.9%) | 65 (24.6%) | 45 (17.0%) | 36 (13.6%) | 54 (20.5%) | 0.010 |
| Vasopressin | 90 (8.5%) | 31 (11.7%) | 16 (6.1%) | 16 (6.1%) | 27 (10.2%) | 0.035 |
| **Outcomes** | | | | | | |
| Length of hospital stay, days | 12 (7, 20) | 12 (6, 23) | 11 (6, 21) | 11 (7, 18) | 12 (7, 20) | 0.492 |
| Length of ICU stay, days | 7 (3, 14) | 7 (3, 13) | 7 (3, 14) | 7 (3, 13) | 8 (3, 15) | 0.155 |
| Hospital mortality, n (%) | 196 (18.6%) | 88 (33.3%) | 38 (14.4%) | 34 (12.9%) | 36 (13.6%) | <0.001 |
| ICU mortality, n (%) | 156 (14.8%) | 68 (25.8%) | 30 (11.4%) | 27 (10.2%) | 31 (11.7%) | <0.001 |

Median (IQR); n (%)Kruskal-Wallis rank sum test; Pearson's Chi-squared test; Fisher's exact test.

ICU, Intensive care unit; WBC, white blood cell; RBC, red Blood cell; RDW, red cell distribution width; PT, prothrombin time; APTT, activated partial thromboplastin time; HR, heart rate; SBP, systolic blood pressure; DBP, diastolic blood pressure; MBP, mean arterial pressure; RR, respiratory rate; SpO$_2$, percutaneous oxygen saturation; APTT, activated partial thromboplastin time; SAPS II, Simplified acute physiology score II; GCS, Glasgow coma score; WFNS, World Federation of Neurosurgical Societies; PRR, platelet/ red cell distribution width.

higher proportion of males among non-survivors (51.5% vs. 41.4%, p = 0.010), and racial disparities were evident, with a larger percentage of non-white ethnicity among non-survivors (p < 0.001).

## The association between baseline PRR and all-cause mortality

The univariate analysis demonstrated that in addition to baseline PRR, age, race, heart failure, myocardial infarction, chronic kidney disease, cirrhosis, coiling, HR, RR, temperature, WBC, sodium, potassium, glucose, AG, PT, INR, urea nitrogen, creatinine, dobutamine, dopamine, SAPS II, and Charlson comorbidity index were all associated with ICU and in-hospital mortalities (all p < 0.05). Whereas the association for the GCS approached significance in ICU mortality (S4 Table).

Table 2 shows an unadjusted and a multivariable-adjusted association between PRR and ICU and in-hospital mortalities. In model I, age, sex, and race variables were adjusted, while model II further adjusted other variables, including

**Table 2. Multivariable cox regression models evaluating the association between PRR and ICU and hospital all-cause mortality.**

| Variable | Crude | | Model I | | Model II | |
|---|---|---|---|---|---|---|
| | HR (95% CI) | p-value | HR (95% CI) | p-value | HR (95% CI) | p-value |
| Hospital all-cause mortality | | | | | | |
| Q1(≤12.67) | 1 (Reference) | | 1 (Reference) | | 1 (Reference) | |
| Q2(12.68–15.99) | 0.46 (0.31, 0.67) | <0.001 | 0.49 (0.34, 0.72) | <0.001 | 0.61 (0.40, 0.92) | 0.017 |
| Q3(16.00–19.41) | 0.44 (0.29, 0.65) | <0.001 | 0.49 (0.33, 0.74) | <0.001 | 0.60 (0.39, 0.92) | 0.020 |
| Q4(≥19.42) | 0.43 (0.29, 0.64) | <0.001 | 0.58 (0.38, 0.86) | 0.007 | 0.60 (0.39, 0.92) | 0.019 |
| P for trend | | <0.001 | | 0.001 | | 0.015 |
| ICU all-cause mortality | | | | | | |
| Q1(≤12.67) | 1 (Reference) | | 1 (Reference) | | 1 (Reference) | |
| Q2(12.68–15.99) | 0.40 (0.26, 0.62) | <0.001 | 0.42 (0.27, 0.64) | <0.001 | 0.54 (0.34, 0.86) | 0.009 |
| Q3(16.00–19.41) | 0.40 (0.26, 0.63) | <0.001 | 0.43 (0.28, 0.68) | <0.001 | 0.55 (0.34, 0.90) | 0.016 |
| Q4(≥19.42) | 0.40 (0.26, 0.61) | <0.001 | 0.51 (0.33, 0.79) | 0.002 | 0.54 (0.34, 0.87) | 0.010 |
| P for trend | | <0.001 | | <0.001 | | 0.010 |

Crude model: adjusted for none.

Model I: adjusted for Age, Gender, and Race.

Model II: adjusted for Age, Gender, Race, Heart failure, Malignant tumor, Chronic kidney disease, Cirrhosis, Pneumonia, WBC, Ureanitrogen, Creatinine, Coiling, Dobutamine, Norepinephrine, Sepsis, and Charlson comorbidity index.

WBC, white blood cell; PRR, platelet/ red cell distribution width; HR, hazard ratio; CI, confidence interval.

hypertension, diabetes, heart failure, myocardial infarction, malignant tumor, chronic kidney disease, cirrhosis, pneumonia, hyperlipoidemia, WBC, RBC, hemoglobin, glucose, anion gap, APTT, urea nitrogen, creatinine, coiling, ventilation, heart rate, SBP, DBP, RR, temperature, dobutamine, dopamine, epinephrine, norepinephrine, vasopressin, sepsis, SAPS II, GCS, WFNS grade, and Charlson comorbidity index. When PRR was used as a continuous variable, the results showed that PRR was associated with ICU mortality (crude model: HR = 0.93, 95% CI: 0.90–0.95, p < 0.001; Model I: HR = 0.95, 95% CI: 0.93–0.98, p < 0.001; Model II: HR = 0.97, 95% CI: 0.94–1.00, p = 0.074), and in-hospital mortality (crude model: HR = 0.94, 95% CI: 0.92–0.96, p < 0.001; Model I: HR = 0.96, 95% CI: 0.94–0.99, p = 0.005; Model II: HR = 0.97, 95% CI: 0.94–0.99, p = 0.043). PRR levels were evaluated as a categorical variable for ICU and hospital all-cause mortality. Q1 (≤12.67) PRR levels were used as the reference. Patients with lower PRR levels had significantly increased in-hospital mortality and ICU mortality. In the crude model, the HR values for ICU mortality in Q2 (12.68–15.99), Q3 (16.00–19.41), and Q4 (≥19.42) were 0.40 (95% CI: 0.26–0.62, p < 0.001), 0.40 (95% CI: 0.26–0.63, p < 0.001), and 0.43 (95% CI: 0.26–0.61, p < 0.001), and for in-hospital mortality were 0.46 (95% CI: 0.31–0.67, p < 0.001), 0.44 (95% CI: 0.29–0.65, p < 0.001), and 0.43 (95% CI: 0.29–0.64, p < 0.001). In Model I, the adjusted HR values for ICU mortality in Q2, Q3, and Q4 were 0.48 (95% CI: 0.30–0.77, p = 0.002), 0.49 (95% CI: 0.31–0.79, p = 0.003), and 0.51 (95% CI: 0.32–0.82, p = 0.005), and for in-hospital mortality were 0.56 (95% CI: 0.37–0.84, p = 0.005), 0.56 (95% CI: 0.37–0.86, p = 0.008), and 0.59 (95% CI: 0.38–0.90, p = 0.014). In addition, in Model II, the adjusted HR values for ICU mortality in Q2, Q3, and Q4 were 0.58 (95% CI: 0.35–0.95, p = 0.032), 0.58 (95% CI: 0.34–0.98, p = 0.043), and 0.47 (95% CI: 0.28–0.81, p = 0.006) and for in-hospital mortality were 0.62 (95% CI: 0.40–0.95, p = 0.028), 0.58 (95% CI: 0.37–0.93, p = 0.022), and 0.50 (95% CI: 0.31–0.80, p = 0.004). Moreover, the K-M curves contrasting the four groups are displayed in Fig 2. The figure indicates that the ICU and hospital survival rates of group Q1 were lower than those of groups Q2, Q3, and Q4 (p < 0.0001).

### Analysis of the non-linear relationship between the baseline PRR and ICU and in-hospital mortality

Possible nonlinear relationships between the change in PRR and ICU and hospital all-cause mortalities were examined by RCSs. RCSs revealed a non-linear connection between PRR and ICU all-cause mortality, which was consistent with

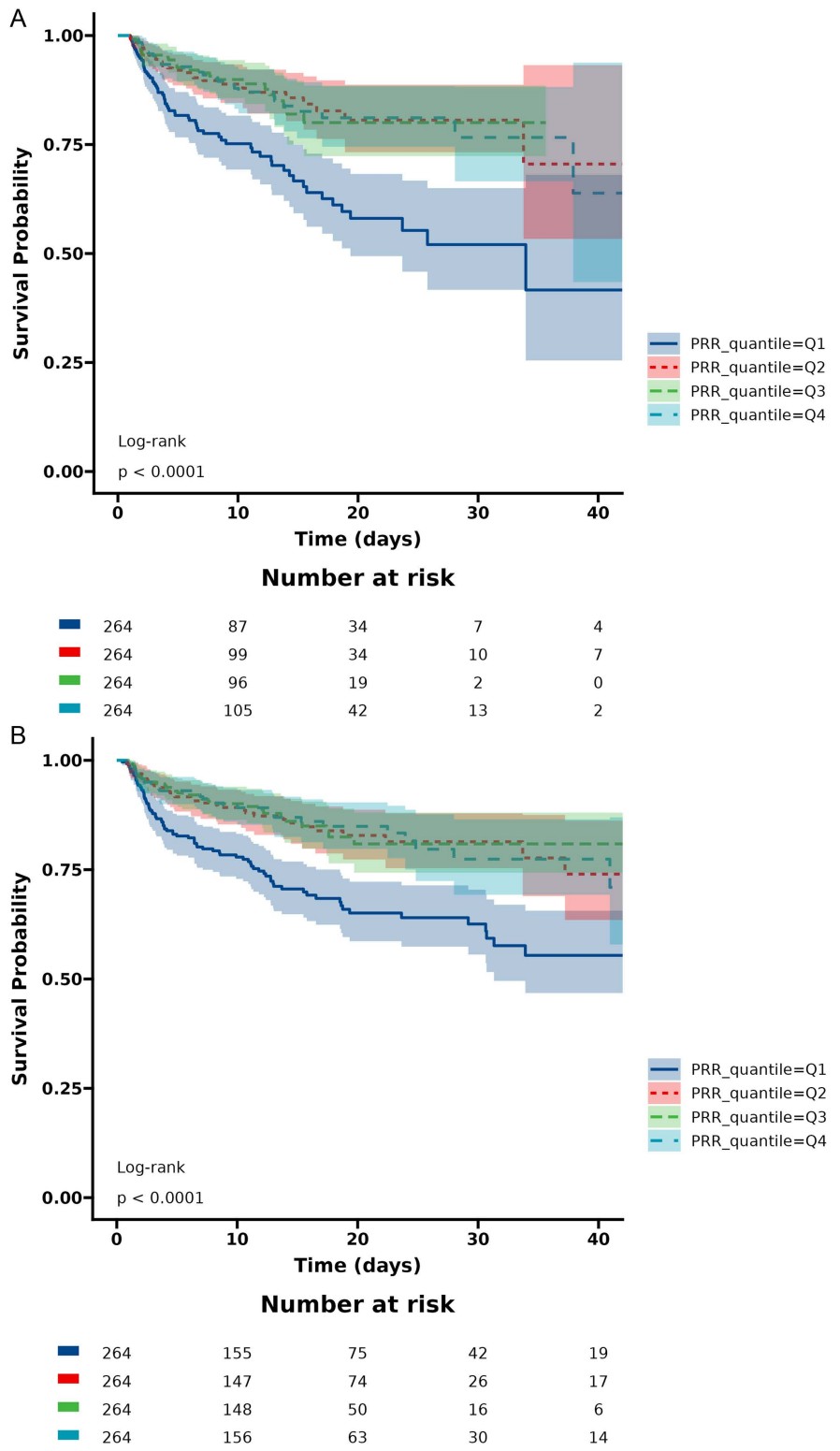

**Fig 2. Kaplan-Meier survival curves for critically ill patients with non-traumatic SAH based on the baseline PRR level.** (A) ICU mortality; (B) In-hospital mortality. X-Axis: survival time (days). Y-Axis: survival probability. PRR, platelet/red cell distribution width; SAH, subarachnoid hemorrhage.

hospital all-cause mortality in participants with non-traumatic SAH. In the threshold analysis, we used a segmented Cox regression to fit the link between the baseline PRR level and ICU and in-hospital mortalities.

The breakpoint of the RCS curve was identified at PRR = 22.6, representing an inflection point in the relationship between the PRR and the ICU mortalities (Fig 3A). The standard Cox regression model represented PRR association with ICU mortality, and HR was 0.93 (95%CI: 0.90–0.95, p < 0.001). Using the breakpoint, the data were stratified into two groups: PRR < 22.6 and PRR ≥ 22.6. The fixed-point piecewise Cox regression was then performed on each group separately. On the left side of the breakpoint, the HR of PRR was 0.91 (95%CI: 0.88–0.94, p < 0.001). This meant that the risk of in-hospital mortality was reduced by 9% per 1 unit increase. On the right side of the inflection point, the HR was 1.03 (95%CI: 0.97–1.10, p = 0.312). It suggested that the association between PRR and ICU mortalities was not statistically significant when the level of PRR was more than 22.6. This meant that the risk of being in the ICU no longer decreased with increasing PRR. The LRT-determined piecewise model provided a significantly better fit than the standard Cox regression model (p = 0.008) (Table 3).

When the breakpoint was 22.6, the standard Cox regression model represented PRR association with in-hospital mortality, and HR was 0.94 (95%CI: 0.92–0.96, p < 0.001) (Fig 3B). The fixed-point piecewise Cox regression was then performed on each group separately. On the left side of the inflection point, the HR of PRR was 0.93 (95% CI: 0.90–0.95, p < 0.001). This meant that the risk of in-hospital mortality was reduced by 8% per 1 unit increase. On the right side of the inflection point, the HR was 1.01 (95% CI: 0.95–1.08, p = 0.693). It suggested that the association between PRR and in-hospital mortality was not statistically significant when the level of PRR was more than 22.6. This meant that the risk of being in hospital no longer decreased with increasing PRR. This meant that the risk of being in the ICU no longer decreased with increasing PRR. The LRT-determined piecewise model provided a significantly better fit than the standard Cox regression model (p = 0.048) (Table 3).

## Subgroup analysis

The subgroup analysis was conducted to reveal the correlation between PRR and ICU and in-hospital mortalities across age (<60 and ≥60 years old), gender, race, hypertension, diabetes, heart failure, sepsis, coiling of aneurysm, SAPS II (<45 and ≥45) GCS (<9 and ≥9), and the results are shown in Fig 4. We found that only sepsis revealed differences in the subgroup analysis between the admission PRR and ICU mortality and hospital mortality, p = 0.033, 0.017, respectively

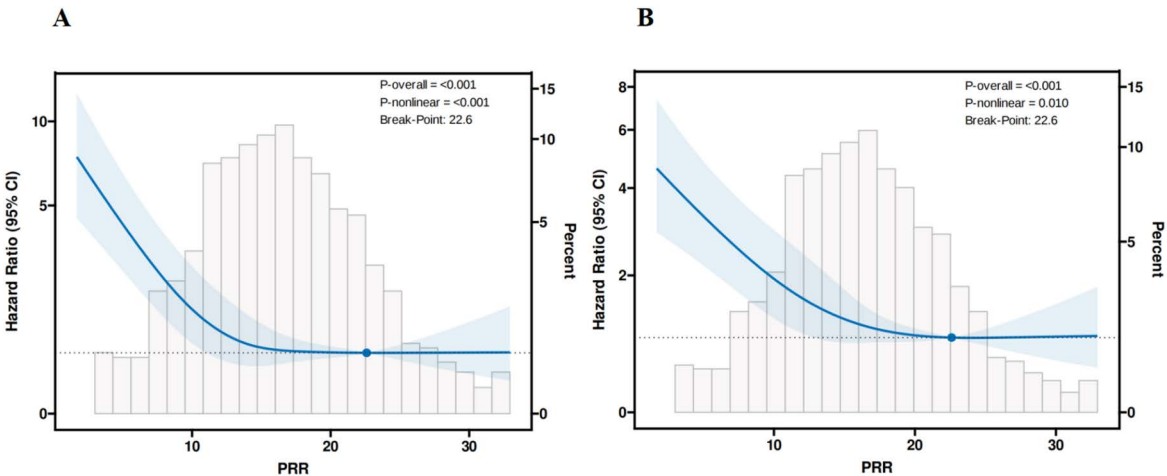

**Fig 3. Non-linear relationship observed between (A) ICU mortality and (B) in-hospital mortality and the baseline PRR level, and the slope changes evidently, which may have a threshold saturation effect.**

**Table 3. Threshold effect analysis of baseline PRR level on ICU mortality and in-hospital mortality in non-traumatic SAH patients.**

| ICU mortality | HR (95% CI) | P-value |
|---|---|---|
| **Fitting by standard Cox regression model** | | |
| Total | 0.93 (0.90, 0.95) | <0.001 |
| **Fitting by piecewise Cox regression model (break-point = 22.6)** | | |
| PRR < 22.6 | 0.91 (0.88, 0.94) | <0.001 |
| PRR ≥ 22.6 | 1.03 (0.97, 1.10) | 0.312 |
| **Log likelihood ratio** | | 0.008 |

| In-hospital mortality | HR (95% CI) | P-value |
|---|---|---|
| **Fitting by standard Cox regression model** | | |
| Total | 0.94 (0.92, 0.96) | <0.001 |
| **Fitting by piecewise Cox regression model (break-point = 22.6)** | | |
| PRR < 22.6 | 0.93 (0.90, 0.95) | <0.001 |
| PRR ≥ 22.6 | 1.01 (0.95, 1.08) | 0.693 |
| **Log likelihood ratio** | | 0.048 |

PRR, platelet/ red cell distribution width; ICU, intensive care unit; SAH, subarachnoid hemorrhage; HR, hazard ratio; CI, confidence interval.

**A**

| Subgroup | N | HR (95% CI) | P for interaction |
|---|---|---|---|
| Overall | 1056 | 0.93 (0.90, 0.95) | |
| Age_group | | | 0.502 |
| < 60 | 413 | 0.92 (0.87, 0.97) | |
| ≥ 60 | 643 | 0.94 (0.91, 0.97) | |
| Gender | | | 0.434 |
| Female | 599 | 0.92 (0.88, 0.95) | |
| Male | 457 | 0.94 (0.90, 0.98) | |
| Race | | | 0.105 |
| White | 609 | 0.90 (0.86, 0.94) | |
| Asian | 43 | 0.96 (0.85, 1.07) | |
| Black | 74 | 0.85 (0.74, 0.99) | |
| Other | 330 | 0.96 (0.92, 1.00) | |
| Hypertension | | | 0.127 |
| No | 521 | 0.91 (0.88, 0.94) | |
| Yes | 535 | 0.95 (0.91, 0.99) | |
| Diabetes | | | 0.167 |
| No | 840 | 0.92 (0.89, 0.95) | |
| Yes | 216 | 0.96 (0.91, 1.02) | |
| Heart_failure | | | 0.233 |
| No | 981 | 0.93 (0.90, 0.95) | |
| Yes | 75 | 0.97 (0.90, 1.05) | |
| Coiling | | | 0.859 |
| No | 872 | 0.93 (0.90, 0.96) | |
| Yes | 184 | 0.94 (0.88, 1.01) | |
| Sepsis | | | 0.033 |
| No | 546 | 0.88 (0.83, 0.93) | |
| Yes | 510 | 0.94 (0.91, 0.97) | |
| SAPSII_group | | | 0.748 |
| < 45 | 934 | 0.95 (0.91, 0.98) | |
| ≥ 45 | 122 | 0.94 (0.91, 0.98) | |
| GCS_group | | | 0.623 |
| < 9 | 143 | 0.92 (0.87, 0.98) | |
| ≥ 9 | 913 | 0.93 (0.90, 0.96) | |

**B**

| Subgroup | N | HR (95% CI) | P for interaction |
|---|---|---|---|
| Overall | 1056 | 0.94 (0.92, 0.96) | |
| Age_group | | | 0.818 |
| < 60 | 413 | 0.94 (0.89, 0.98) | |
| ≥ 60 | 643 | 0.94 (0.92, 0.97) | |
| Gender | | | 0.315 |
| Female | 599 | 0.93 (0.90, 0.96) | |
| Male | 457 | 0.95 (0.92, 0.99) | |
| Race | | | 0.005 |
| White | 609 | 0.92 (0.89, 0.95) | |
| Asian | 43 | 0.98 (0.88, 1.09) | |
| Black | 74 | 0.82 (0.71, 0.93) | |
| Other | 330 | 0.98 (0.94, 1.01) | |
| Hypertension | | | 0.212 |
| No | 521 | 0.93 (0.90, 0.96) | |
| Yes | 535 | 0.96 (0.92, 0.99) | |
| Diabetes | | | 0.322 |
| No | 840 | 0.93 (0.91, 0.96) | |
| Yes | 216 | 0.97 (0.92, 1.01) | |
| Heart_failure | | | 0.158 |
| No | 981 | 0.94 (0.91, 0.96) | |
| Yes | 75 | 0.99 (0.92, 1.06) | |
| Coiling | | | 0.858 |
| No | 872 | 0.94 (0.92, 0.97) | |
| Yes | 184 | 0.94 (0.88, 1.01) | |
| Sepsis | | | 0.017 |
| No | 546 | 0.90 (0.85, 0.95) | |
| Yes | 510 | 0.96 (0.93, 0.98) | |
| SAPSII_group | | | 0.317 |
| < 45 | 934 | 0.95 (0.92, 0.97) | |
| ≥ 45 | 122 | 0.97 (0.93, 1.00) | |
| GCS_group | | | 0.901 |
| < 9 | 143 | 0.94 (0.89, 0.99) | |
| ≥ 9 | 913 | 0.94 (0.92, 0.97) | |

**Fig 4. The relationship between PRR and (A) ICU mortality and (B) in-hospital mortality in subgroup analysis.**

(Table 4). The interaction between the PRR and other subgroup factors was analyzed, and significant interactions were not observed (p for interaction > 0.05). Sensitivity analysis showed that the interaction was not significant in different subgroups, which proved the robustness of PRR threshold (p for interaction > 0.05) (S5 and S6 Tables).

## Discuccion

Our study presents several novel and significant findings regarding the PRR in non-traumatic SAH. Firstly, we identified PRR as a potential biomarker for non-traumatic SAH, pioneering the exploration of this combined ratio in relation to

**Table 4. The subgroup analysis for baseline PRR on ICU and hospital mortality.**

| Subgroup | N | ICU mortality | | In-hospital mortality | |
|---|---|---|---|---|---|
| | | HR (95% CI) | P for interaction | HR (95% CI) | P for interaction |
| Overall | 1056 | 0.93 (0.90-0.95) | | 0.94 (0.92-0.96) | |
| Age | | | 0.502 | | 0.818 |
| < 60 | 413 | 0.92 (0.87-0.97) | | 0.94 (0.89-0.98) | |
| ≥ 60 | 643 | 0.94 (0.91-0.97) | | 0.94 (0.92-0.97) | |
| Gender | | | 0.434 | | 0.315 |
| Female | 599 | 0.92 (0.88-0.95) | | 0.93 (0.90-0.96) | |
| Male | 457 | 0.94 (0.90-0.98) | | 0.95 (0.92-0.99) | |
| Race | | | 0.105 | | 0.005 |
| White | 609 | 0.90 (0.86-0.94) | | 0.92 (0.89-0.95) | |
| Asian | 43 | 0.96 (0.85-1.07) | | 0.98 (0.88-1.09) | |
| Black | 74 | 0.85 (0.74-0.99) | | 0.82 (0.71-0.93) | |
| Other | 330 | 0.96 (0.92-1.00) | | 0.98 (0.94-1.01) | |
| Hypertension | | | 0.127 | | 0.212 |
| No | 521 | 0.91 (0.88-0.94) | | 0.93 (0.90-0.96) | |
| Yes | 535 | 0.95 (0.91-0.99) | | 0.96 (0.92-0.99) | |
| Diabetes | | | 0.167 | | 0.322 |
| No | 840 | 0.92 (0.89-0.95) | | 0.93 (0.91-0.96) | |
| Yes | 216 | 0.96 (0.91-1.02) | | 0.97 (0.92-1.01) | |
| Heart failure | | | 0.233 | | 0.158 |
| No | 981 | 0.93 (0.90-0.95) | | 0.94 (0.91-0.96) | |
| Yes | 75 | 0.97 (0.90-1.05) | | 0.99 (0.92-1.06) | |
| Coiling | | | 0.859 | | 0.858 |
| No | 872 | 0.93 (0.90-0.96) | | 0.94 (0.92-0.97) | |
| Yes | 184 | 0.94 (0.88-1.01) | | 0.94 (0.88-1.01) | |
| Sepsis | | | 0.033 | | 0.017 |
| No | 546 | 0.88 (0.83-0.93) | | 0.90 (0.85-0.95) | |
| Yes | 510 | 0.94 (0.91-0.97) | | 0.96 (0.93-0.98) | |
| SAPS II | | | 0.748 | | 0.317 |
| < 45 | 934 | 0.95 (0.91-0.98) | | 0.95 (0.92-0.97) | |
| ≥ 45 | 122 | 0.94 (0.91-0.98) | | 0.97 (0.93-1.00) | |
| GCS | | | 0.623 | | 0.901 |
| < 9 | 143 | 0.92 (0.87-0.98) | | 0.94 (0.89-0.99) | |
| ≥ 9 | 913 | 0.93 (0.90-0.96) | | 0.94 (0.92-0.97) | |

PRR, platelet/ red cell distribution width; ICU, intensive care unit; SAPS II, simplified acute physiology score; GCS, glasgow coma scale; HR, hazard ratio; CI, confidence interval.

patient outcomes, unlike prior studies that focused on its individual components separately. This offers fresh insights into SAH pathophysiology, highlighting the complex and crucial interplay between platelet- and red blood cell-related factors. We discovered a significant inverse association between admission PRR and the risk of ICU and in-hospital mortality in non-traumatic SAH patients. Patients with low PRR levels had higher mortality and shorter survival time. In addition, there was a nonlinear correlation between PRR and all-cause mortality at admission. Notably, the threshold effect observed at PRR = 22.6 suggests a dichotomous prognostic role: below this critical value, each unit increase in PRR correlates with reduced all-cause mortality risk, whereas no such association exists above the threshold. This nonlinear pattern may reflect dual pathophysiological mechanisms. In the subgroup analysis, there was no significant interaction between admission PRR level and race in ICU mortality, but a significant one in hospitalization mortality. We suspect that the small number of Asian participants affects the results. Also, comorbidities in certain racial groups that affect platelet or red blood cell function may confound the association between PRR and SAH outcomes, yet current evidence can't prove this. Sensitivity analysis showed that the interaction between different subgroups was not significant (p value of interaction > 0.05), except for gender and race. This is because the population distribution has changed after threshold grouping, and prospective cohort study should be conducted in the future to further confirm its consistency. However, on the whole, the threshold of admission PRR (22.6) is robust in different subgroups. From the aspect of platelet function, the change of platelet quantity or activity around the threshold may have an influence on coagulation function and inflammatory reaction; From the point of view of RBC fragility, the stability of erythrocytes at different PRR levels is inconsistent and affects the oxygen transport function.

While previous studies have independently linked thrombocytopenia [5] or elevated RDW [6−9] to adverse outcomes in stroke cohorts, our demonstration of PRR's non-linear prognostic power underscores the critical synergy between these parameters. Unlike the linear platelet count-mortality association reported in ischemic stroke (OR=1.19 for platelet<150 × 10⁹/L), the threshold effect observed here (PRR = 22.6) suggests a pathophysiological tipping point in SAH. We posit that PRR < 22.6 reflects concurrent platelet exhaustion and erythrocyte destabilization—processes amplified by SAH-specific mechanisms such as cerebrospinal fluid thrombin surge [10, 11] and hemoglobin-derived oxidative stress [12, 13]. This integrative biomarker approach addresses a key limitation of conventional univariate analyses, which overlook compensatory interactions between coagulation and erythropoiesis systems. The differential prognostic value of PRR in non-traumatic SAH versus other neurological emergencies merits emphasis. This divergence likely stems from SAH's unique thromboinflammatory cascade: non-traumatic SAH induces PAR-1-mediated platelet apoptosis [14] and TLR4-dependent eryptosis [15], processes captured synergistically by PRR but absent in trauma-dominated pathophysiology [16]. PRR's operational simplicity confers practical advantages over complex indices requiring specialized assays. PRR utilizes routinely available complete blood count parameters, enabling real-time risk stratification in emergency settings. Moreover, the stability of PRR-mortality association across subgroups (age, GCS group) contrasts with dynamic platelet monitoring strategies, which demand serial measurements. These features position PRR as a pragmatic triage tool, particularly valuable in resource-constrained environments. Post-SAH, thrombin and matrix metalloproteinase-9 (MMP-9) surge in cerebrospinal fluid induce platelet hyperactivation and apoptosis via protease-activated receptor 1 (PAR-1) signaling [14, 17, 18]. This pathological platelet depletion reduces circulating platelet counts, while platelet-derived microparticles (PMPs) exacerbate neurovascular inflammation by breaching the blood-brain barrier [19]. Elevated RDW transcends erythrocyte production imbalance, correlating strongly with SAH-induced systemic oxidative stress. Preclinical models demonstrate that subarachnoid hemoglobin derivatives activate TLR4/NF-κB pathways, triggering pro-inflammatory eryptosis and increasing immature erythrocyte proportions [20]. Concurrently, SAH-associated sympathetic overactivation suppresses bone marrow erythropoietin responsiveness via β-adrenergic receptors, amplifying RDW variability [21]. The synergistic interplay between rising RDW and platelet counts creates a vicious cycle. Enhanced CD40L-PSGL-1 interactions on platelet-RBC membranes post-SAH drive neutrophil extracellular trap (NET) formation [22]. Low PRR levels signify platelet-mediated immunoregulatory failure, permitting NET overproduction that induces cerebral DNA damage and

blood-brain barrier disruption [23]. Concurrently, erythrocyte debris activates microglia via the C3a-C5a complement axis, amplifying IL-6/IL-1β storms [24]. Studies have revealed that suppressed mitochondrial complex activity in brain endothelial cells of SAH patients, with compensatory glycolytic upregulation [25]. Depleted platelet-derived serotonin (5-HT) impairs vasomotor regulation, while RDW-driven hyperviscosity compromises cerebral oxygen extraction. This dual insult reaches criticality at PRR < 22.6, triggering irreversible energetic failure. At PRR ≥ 22.6, adequate platelet reserves sustain vascular integrity and clear apoptotic erythrocytes, while moderate RDW elevation may indicate adaptive stress erythropoiesis. Below this threshold, disrupted platelet-RBC homeostasis unleashes thromboinflammatory cascades. Isolated interventions (e.g., platelet transfusion) may fail unless synchronized with erythrocyte dynamics modulation [26].

Our research presents the following distinct advantages: (1) To the best of our knowledge, this is the inaugural study delving into the correlation between the baseline levels of the parameter PRR in patients with non-traumatic SAH and their mortality rates during the ICU stay and throughout hospitalization. By exploring this relationship, we aim to fill a significant gap in the existing literature and offer novel insights into the prognostic factors for SAH patients. (2) Leveraging real-world data, our study is designed to encompass a large-scale and diverse population. The remarkably low proportion of missing values for PRR is a notable strength. This characteristic is expected to effectively mitigate selection bias, ensuring that the results are more representative of the general patient population and enhancing the reliability and generalizability of our findings. (3) In the data analysis process, we adopted a two-segment Cox proportional hazards regression model. This advanced statistical approach enabled us to comprehensively analyze the threshold effect of the relationship between PRR and all-cause mortality. By doing so, we can not only identify the potential turning points in the relationship but also provide more accurate and nuanced risk assessments for SAH patients.

Several limitations warrant acknowledgment. First, the retrospective MIMIC-IV design precludes assessment of PRR dynamics post-admission—a critical gap given Chardon et al.'s (2024) findings on platelet trajectory prognostic value [27]. Because the dynamic change information of PRR can not be obtained, there may be some limitations in clinical practice, such as whether to carry out platelet transfusion and when to carry out treatment. Future prospective trials should evaluate whether PRR-guided interventions (e.g., early platelet transfusion for PRR < 22.6) improve outcomes. Second, unmeasured confounders may influence observed associations. For example, the specific location of aneurysm characteristics (location, size and shape) and the potential impact of intervention timing (treatment at different times after bleeding) on the research results. Third, external validation across ethnic populations is essential, as RDW cutoffs vary with genetic and nutritional factors.

## Conclusion

There was a non-linear connection between the baseline PRR level and in-hospital mortality. A low level of PRR could increase the risk of death in participants with non-traumatic SAH.

## Supporting information

**S1 Table. Details of missing values.**
(DOCX)

**S2 Table. Variance Inflation Factor and Tolerance.**
(DOCX)

**S3 Table. Patient demographics and baseline characteristics.**
(DOCX)

**S4 Table. Univariate Cox regression analyses for ICU and in-hospital mortality in patients with non-traumatic SAH.**
(DOCX)

**S5 Table. Sensitivity analysis: subgroup analysis of PRR threshold [ICU mortality].**
(DOCX)

**S6 Table. Sensitivity analysis: subgroup analysis of PRR threshold [Hospital mortality].**
(DOCX)

## Author contributions

**Conceptualization:** Yang Liu, Guang Feng.

**Data curation:** Yang Liu, Yi Shi, Pengzhao Zhang, Mengyuan Xu, Jiaqi Zhang, Jing Xia, Shaojie Guo, Gaofeng Li, Guang Feng.

**Formal analysis:** Yang Liu.

**Funding acquisition:** Guang Feng.

**Supervision:** Guang Feng.

**Writing – original draft:** Yang Liu.

**Writing – review & editing:** Yang Liu, Guang Feng.

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
