## [Decision Letter · Decision Letter 0]

6 May 2025

PONE-D-25-11821Relationship between baseline platelet-to-red blood cell distribution width ratio and all-cause mortality in non-traumatic subarachnoid hemorrhage: a retrospective analysis of the MIMIC-IV databasePLOS ONE

Dear Dr. Feng,

Thank you for submitting your manuscript to PLOS ONE. After careful consideration, we feel that it has merit but does not fully meet PLOS ONE’s publication criteria as it currently stands. Therefore, we invite you to submit a revised version of the manuscript that addresses the points raised during the review process.

We look forward to receiving your revised manuscript.

Kind regards,

Yimin Chen

Academic Editor

PLOS ONE

Journal Requirements:

Health Commission of Henan Province (grant numbers: SBGJ202402008 ).

3. Your abstract cannot contain citations. Please only include citations in the body text of the manuscript, and ensure that they remain in ascending numerical order on first mention.

4. Please remove all personal information, ensure that the data shared are in accordance with participant consent, and re-upload a fully anonymized data set.

Additional guidance on preparing raw data for publication can be found in our Data Policy (https://journals.plos.org/plosone/s/data-availability#loc-human-research-participant-data-and-other-sensitive-data) and in the following article:http://www.bmj.com/content/340/bmj.c181.long.

Reviewers' comments:

Reviewer's Responses to Questions

**Comments to the Author**

1. Is the manuscript technically sound, and do the data support the conclusions?

Reviewer #1: Yes

Reviewer #2: Yes

2. Has the statistical analysis been performed appropriately and rigorously? 

Reviewer #1: Yes

Reviewer #2: Yes

3. Have the authors made all data underlying the findings in their manuscript fully available?

Reviewer #1: Yes

Reviewer #2: Yes

4. Is the manuscript presented in an intelligible fashion and written in standard English?

Reviewer #1: Yes

Reviewer #2: Yes

5. Review Comments to the Author

Reviewer #1: In this study, the authors performed a comprehensive analysis of the association between PRR levels and outcomes in patients with non-traumatic subarachnoid hemorrhage. The presentation of the Results section was detailed, and the findings of the exploration had certain clinical value. In this regard, I would like to give some comments for the author's consideration.

1. In the Introduction, the author described, "Early brain injury, which occurs within the first 72 h after SAH, is characterized by a cascade of inflammatory responses, oxidative stress, and neuronal apoptosis. Targeting these pathological mechanisms may improve outcomes and reduce mortality in patients with non-traumatic SAH." The authors should further elaborate on the potential association between PRR and these pathophysiological mechanisms, which would provide more referral for the presentation of PRR.

2. In the Methods, the authors included all baseline information in the regression analysis as a potential confounding factor, which may result in the greatest reduction in confounding bias. However, the presence of multicollinearity can distort the real effect. Therefore, the authors need to perform multicollinearity analyses to exclude variables that should not be included in the regression analysis.

3. In figure4, the abscissa range of subgroup analyses should be narrowed, not from zero to two. This will make it more aesthetically pleasing.

4. The authors discussed in detail the possible mechanisms of the association between PRR and outcomes, and cited some of the advantages of this study. However, on this basis, the authors should discuss the novel findings of this study.

Reviewer #2: Reviewer Comments

Overall Impression and Recommendation

This retrospective cohort study investigates the association between baseline platelet-to-red blood cell distribution width ratio (PRR) and mortality in non-traumatic subarachnoid hemorrhage (SAH) using the MIMIC-IV database. The study addresses an important clinical question and provides novel insights into PRR’s prognostic utility. However, significant limitations in design, interpretation, and reporting necessitate "major revisions" before publication.

Strengths

1. Clinical Relevance: The study fills a gap in understanding the prognostic role of PRR in non-traumatic SAH, a condition with high morbidity/mortality.

2. Large Dataset: Use of the MIMIC-IV database enhances generalizability, and the sample size (n=1,056) supports subgroup analyses.

3. Statistical Rigor: Application of Cox regression, Kaplan-Meier curves, and restricted cubic spline (RCS) analysis to explore non-linear relationships is appropriate.

4. Threshold Effect: Identification of a PRR threshold (22.6) adds nuance to risk stratification, suggesting a potential clinical cut-off.

Key Concerns and Weaknesses

1. Retrospective Design:

- Observational nature limits causal inference; unmeasured confounders (e.g., aneurysm location, treatment details) may influence outcomes.

- Dynamic PRR changes post-admission are not evaluated, despite prior evidence linking platelet dynamics to SAH outcomes (e.g., Chardon et al., 2024).

2. Generalizability:

- MIMIC-IV data are from a single U.S. institution; external validation across diverse populations is required.

- Racial/ethnic subgroup analyses are underpowered, limiting conclusions about disparities.

3. Methodological Limitations:

- Exclusion of patients with ICU stays <24 hours may introduce selection bias.

- Missing data handling (e.g., platelet/RDW values) is not explicitly described.

- Subgroup interactions (e.g., age, WFNS grade) lack sufficient power for definitive conclusions.

4. Threshold Interpretation:

- The PRR threshold (22.6) requires replication in independent cohorts and mechanistic validation to justify clinical utility.

- The RCS analysis’s biological plausibility at this threshold is not discussed.

5. Mechanistic Insight:

- The proposed pathophysiological pathways (e.g., platelet-RBC interplay) are speculative and not directly supported by experimental data.

Specific Recommendations for Revision

1. Expand Discussion on Limitations:

- Acknowledge the retrospective design’s inability to establish causality and the potential impact of unmeasured confounders (e.g., aneurysm characteristics, timing of intervention).

- Discuss the lack of dynamic PRR data and its implications for clinical practice.

2. Enhance Transparency in Methods:

- Clarify how missing platelet/RDW data were handled (e.g., exclusion criteria, imputation methods).

- Justify the exclusion of patients with ICU stays <24 hours and assess sensitivity to this criterion.

3. Strengthen Subgroup Analyses:

- Report power calculations for subgroup interactions to contextualize non-significant findings.

- Highlight racial/ethnic disparities in baseline characteristics (e.g., Table 1) and discuss implications.

4. Reinterpret Threshold Findings:

- Conduct sensitivity analyses to test the robustness of the PRR threshold (22.6) across subgroups.

- Provide mechanistic hypotheses for the observed threshold (e.g., platelet function vs. RBC fragility).

5. Address Clinical Implications:

- Acknowledge that PRR-guided interventions (e.g., platelet transfusion) are not validated and require prospective trials.

- Recommend future studies to evaluate PRR dynamics and combined platelet/RBC therapies.

6. Editorial Improvements:

- Correct inconsistencies in notation (e.g., "Q4 (≥19.42)" vs. "Q4 (219.42)" in Results).

- Ensure clarity in statistical model descriptions (e.g., Model II adjustments).

Conclusion

This study offers valuable insights into PRR’s prognostic potential in non-traumatic SAH but requires major revisions to address design limitations, enhance transparency, and strengthen interpretations. The identified threshold (PRR=22.6) warrants further validation, and mechanistic exploration is critical to advancing clinical translation. With careful revisions, this work could contribute significantly to the field.

Final Note: The authors should provide a point-by-point response to these comments and revise the manuscript accordingly. A re-submission with these improvements would be reconsidered for publication.

6. PLOS authors have the option to publish the peer review history of their article (what does this mean? ). If published, this will include your full peer review and any attached files.

**Do you want your identity to be public for this peer review?** For information about this choice, including consent withdrawal, please see our Privacy Policy .

Reviewer #1: **Yes: ** Kai Wang

Reviewer #2: No

---

## [Author Response · Author response to Decision Letter 1]

24 May 2025

Dear Editor and Reviewers,

Many thanks to you for your valuable advice and recognition of our manuscript. We sincerely appreciate the time and expertise you have invested in reviewing our work, as your insightful comments have provided not only essential guidance for revising the paper but also important direction for our research. We have carefully studied each comment and made targeted corrections that we believe address your concerns. Specific locations of modifications in the revised manuscript are detailed in the attached Response to Reviewers document. To facilitate clarity, revised paragraphs are color-coded: blue for suggestions from Reviewer 1 and red for those from Reviewer 2. We have addressed the comments and suggestions point by point.

---

## [Decision Letter · Decision Letter 1]

7 Aug 2025

Relationship between baseline platelet-to-red blood cell distribution width ratio and all-cause mortality in non-traumatic subarachnoid hemorrhage: a retrospective analysis of the MIMIC-IV database

PONE-D-25-11821R1

Dear Dr. Feng,

We’re pleased to inform you that your manuscript has been judged scientifically suitable for publication and will be formally accepted for publication once it meets all outstanding technical requirements.

Kind regards,

Yimin Chen

Academic Editor

PLOS ONE

Additional Editor Comments (optional):

Reviewers' comments:

Reviewer's Responses to Questions

**Comments to the Author**

1. If the authors have adequately addressed your comments raised in a previous round of review and you feel that this manuscript is now acceptable for publication, you may indicate that here to bypass the “Comments to the Author” section, enter your conflict of interest statement in the “Confidential to Editor” section, and submit your "Accept" recommendation.

Reviewer #1: All comments have been addressed

Reviewer #2: All comments have been addressed

2. Is the manuscript technically sound, and do the data support the conclusions?

Reviewer #1: Yes

Reviewer #2: Yes

3. Has the statistical analysis been performed appropriately and rigorously? 

Reviewer #1: Yes

Reviewer #2: Yes

4. Have the authors made all data underlying the findings in their manuscript fully available?

Reviewer #1: Yes

Reviewer #2: Yes

5. Is the manuscript presented in an intelligible fashion and written in standard English?

Reviewer #1: Yes

Reviewer #2: Yes

6. Review Comments to the Author

Reviewer #1: After a thorough revision, the manuscript reached the level of publication. I recommend the acceptance of this manuscript.

Reviewer #2: (No Response)

7. PLOS authors have the option to publish the peer review history of their article (what does this mean? ). If published, this will include your full peer review and any attached files.

**Do you want your identity to be public for this peer review?** For information about this choice, including consent withdrawal, please see our Privacy Policy .

Reviewer #1: **Yes: ** Kai Wang

Reviewer #2: No

---

## [Editor Report · Acceptance letter]

PONE-D-25-11821R1

PLOS ONE

Dear Dr. Feng,

I'm pleased to inform you that your manuscript has been deemed suitable for publication in PLOS ONE. Congratulations! Your manuscript is now being handed over to our production team.

Kind regards,

on behalf of

Dr. Yimin Chen

Academic Editor

PLOS ONE